# LLMCarbon: Modeling the End-To-End Carbon Footprint of Large Language Models[*]

**Ahmad Faiz, Sotaro Kaneda, Ruhan Wang, Rita Osi[†], Prateek Sharma, Fan Chen, Lei Jiang**
Indiana University    [†]Jackson State University
{afaiz,skaneda,ruhwang,prateeks,fc7,jiang60}@iu.edu
[†]j00967039@students.jsums.edu

## Abstract

The carbon footprint associated with large language models (LLMs) is a significant concern, encompassing emissions from their training, inference, experimentation, and storage processes, including operational and embodied carbon emissions. An essential aspect is accurately estimating the carbon impact of emerging LLMs even before their training, which heavily relies on GPU usage. Existing studies have reported the carbon footprint of LLM training, but only one tool, mlco2, can predict the carbon footprint of new neural networks prior to physical training. However, mlco2 has several serious limitations. It cannot extend its estimation to dense or mixture-of-experts (MoE) LLMs, disregards critical architectural parameters, focuses solely on GPUs, and cannot model embodied carbon footprints. Addressing these gaps, we introduce *LLMCarbon*, an end-to-end carbon footprint projection model designed for both dense and MoE LLMs. Compared to mlco2, LLMCarbon significantly enhances the accuracy of carbon footprint estimations for various LLMs. The source code is released at https://github.com/SotaroKaneda/MLCarbon.

## 1 Introduction

Large language models (LLMs) have established their supremacy in addressing a wide spectrum of natural language processing tasks (Brown et al., 2020). However, the proliferation of these models, coupled with increasingly expansive datasets (Sanderson, 2023; Anil et al., 2023), has woven LLM inferences into the fabric of everyday life (Campello de Souza et al., 2023). This surge in LLM adoption has, in turn, exacerbated the already considerable environmental impacts associated with machine learning (ML) (Thompson et al., 2021). For instance, the creation of a transformer with 213 million parameters through neural architecture search has been likened to the carbon dioxide equivalent (CO2eq) emissions of five cars over their entire lifespans (Strubell et al., 2019).

Given the ecological implications of LLMs, it becomes essential for both cloud service providers and regular users to gain a profound understanding of the carbon footprint of emerging LLMs. This awareness is particularly critical before embarking on resource-intensive training endeavors that entail the utilization of thousands of GPUs. During the initial design phase, key parameters such as the LLM's parameter count, hardware configurations, and the energy efficiency of the hosting data center need to be factored into a robust carbon footprint projection model. This model should possess the capability to swiftly and accurately estimate the carbon footprint, encompassing both *operational* and *embodied* carbon emissions. Moreover, it should provide valuable insights into metrics like test loss, training duration, and inference latency, all crucial aspects of LLM performance. The existence of such a carbon footprint projection model empowers cloud providers to intelligently explore the trade-off between test loss and carbon footprint when designing new LLMs. Additionally, it encourages everyday users to adopt practices that mitigate LLM carbon footprints by facilitating quantitative comparisons across various LLM configurations.

Currently, *there is a notable void in the availability of a comprehensive end-to-end carbon footprint projection model tailored specifically for LLMs.* Prior research efforts (Henderson et al., 2020; Wu et al., 2022; Anthony et al., 2020; Schwartz et al., 2020; Patterson et al., 2021; Dodge et al., 2022;

---

[*]This work was supported in part by CCF-2105972, and NSF CAREER AWARD CNS-2143120.

Strubell et al., 2019; Lakim et al., 2022) have predominantly focused on recording and reporting the carbon footprint associated with the training phase of ML models. To date, only one tool, mlco2 (Lacoste et al., 2019), has emerged capable of predicting the carbon footprint of an ML task based on parameters like GPU usage, training duration, and data center efficiency. However, mlco2 exhibits several serious limitations. Firstly, it is confined to convolutional neural networks (CNNs) and cannot extend its estimations to include the carbon footprint of LLMs. Secondly, mlco2 neglects crucial architectural aspects of ML models, such as parameter counts, resulting in overestimated projections. Thirdly, it exclusively considers GPUs, disregarding specialized ML hardware like TPUs (Jouppi et al., 2017), and assumes uniform peak computing throughput across GPUs, leading to inaccuracies in its carbon footprint assessments. Lastly, although the embodied carbon footprint of an ML task holds equal significance to its operational carbon footprint (Wu et al., 2022), mlco2 is incapable of modeling the embodied carbon footprint of an LLM based on its hardware resources.

In this paper, we propose an end-to-end carbon footprint projection model, *LLMCarbon*, which can accurately predict the carbon footprint of both dense and MoE LLMs during their training, inference, experimentation, and storage phases. LLMCarbon incorporates critical LLM, hardware, and data center parameters, such as LLM parameter count, hardware type, system power, chip area, and data center efficiency, to model both operational and embodied carbon footprints of an LLM. When validated against Google's published LLM carbon footprints, the results generated by LLM-Carbon exhibit differences of only $\leq 8.2\%$, and thus are more accurate than those of mlco2.

## 2 BACKGROUND

**LLM Carbon Footprint**. The carbon footprint of a LLM comprises two fundamental components (Gupta et al., 2022): the operational footprint, encompassing emissions stemming from hardware energy consumption, and the embodied footprint, encapsulating emissions arising from hardware manufacturing. Previous investigations (Henderson et al., 2020; Wu et al., 2022; Anthony et al., 2020; Schwartz et al., 2020; Patterson et al., 2022; Dodge et al., 2022; Strubell et al., 2019) have predominantly focused on recording and reporting the operational carbon footprint of various ML tasks. A notable exception is Wu et al. (2022), which delved into the embodied carbon footprint of ML tasks and revealed that within a Meta data center, the embodied carbon footprint of an LLM constitutes $\sim 50\%$ of its operational carbon footprint.

**Neural Scaling Law**. The Neural Scaling Law (Kaplan et al., 2020) delineates a power-law relationship linking an LLM's test loss to three key factors: the number of model parameters, the scale of the training dataset, and the computational resources utilized during training. This relationship holds across diverse architectures and downstream ML tasks, spanning zero-shot, prompted, and fine-tuned scenarios (Caballero et al., 2023).

**Reducing LLM Carbon Footprint**. Efforts on reducing LLM carbon footprints have been channeled into 4 domains. Firstly, sparse MoE architectures (Fedus et al., 2022) have been proposed to enhance LLM performance by increasing model parameters while maintaining a similar computational load. Secondly, the adoption of specialized ML hardware, such as TPUs (Jouppi et al., 2017), has emerged as a more energy-efficient alternative to power-hungry GPUs. Thirdly, ML-focused data centers have optimized their facilities into large-scale systems, reducing cooling and infrastructure overhead to enhance power usage effectiveness (PUE) (Liu et al., 2020). Lastly, these data centers are transitioning to renewable energy sources like solar and wind power (Acun et al., 2023) to mitigate the operational carbon footprint of LLMs. However, the recent proliferation of ML-specific hardware within these data centers, driven by the diverse demands of ML tasks, is widening the gap between operational and embodied carbon footprints in the near future (Wu et al., 2022).

**Parallelism in LLM Processing**. Effective processing of LLMs necessitates the utilization of multiple computing devices, such as GPUs or TPUs, owing to significant LLM parameter counts. Four types of parallelism, i.e., data, tensor, pipeline, and expert, are commonly employed to enhance hardware efficiency, quantified as actual throughput relative to peak throughput.

- **Data Parallelism**: In data parallelism (Xing et al., 2015), the full LLM model is distributed to each computing device, while the input dataset is divided among these devices. Periodic gradient aggregation ensures that all devices maintain consistent model weights.
- **Tensor Parallelism**: Tensor parallelism (Narayanan et al., 2021) involves distributing an LLM's layers across multiple devices. Within a transformer layer, the self-attention block partitions key, query, and value matrices through column-wise division. The output linear layer directly handles

Table 1: The comparison of LLMCarbon against prior work.

| scheme | predictive modeling | MoE support | architectural parameters | specialized hardware | operational carbon | embodied carbon |
|---|---|---|---|---|---|---|
| mlco2 | ✓ | ✗ | ✗ | ✗ | ✓ | ✗ |
| others | ✗ | ✗ | ✗ | ✗ | ✓ | ✓ |
| **LLMCarbon** | ✓ | ✓ | ✓ | ✓ | ✓ | ✓ |

the attention operation's partitioned output, with weight matrix partitioning by rows. In the two-layer MLP, the first layer is divided along columns, and the second along rows. Efficient data coordination among partitions on different devices is achieved through two all-reduce operations in forward and backward passes.

- **Pipeline Parallelism**: In pipeline parallelism (Narayanan et al., 2021), an LLM's layers are distributed across multiple devices. Each device handles an equal number of layers, and microbatches split a batch for pipelined execution. Synchronous weight updates are ensured through pipelining. However, periodic pipeline flushes to synchronize steps across devices introduce "pipeline bubbles" at batch starts and ends, which need to be minimized for efficient pipeline model parallelism.
- **Expert Parallelism**: Expert parallelism (Kim et al., 2021) is tailored for parallelizing the training of MoE LLMs. This approach involves distributing distinct experts across various devices, enabling parallel execution. However, due to the separation of experts across multiple computing devices, explicit communication using all-to-all primitives becomes essential.

## 3 RELATED WORK

Table 1 provides a comparison between LLMCarbon and existing research endeavors. The predominant focus of prior studies (Henderson et al., 2020; Wu et al., 2022; Anthony et al., 2020; Schwartz et al., 2020; Dodge et al., 2022; Strubell et al., 2019) has been the measurement and reporting of carbon footprints associated with the actual training phase of ML models, denoted as "others" in the table. Notably, only one previous model, mlco2 (Lacoste et al., 2019), possesses the capability to predict the carbon footprint of an ML task based on metrics like GPU utilization, training duration, and data center efficiency. Nevertheless, mlco2 encounters four significant limitations. Firstly, mlco2 cannot estimate the carbon footprint of LLMs, particularly sparse MoE LLMs. Secondly, it overlooks essential architectural attributes of LLMs, such as LLM parameter count, resulting in exaggerated predictions. Thirdly, mlco2 exclusively considers GPUs and neglects specialized ML hardware like TPUs (Jouppi et al., 2017), assuming uniform peak computing throughput across all GPUs, thereby yielding imprecise carbon footprint estimations. Lastly, mlco2 cannot model the embodied carbon footprint of an LLM based on its hardware configuration.

## 4 LLMCARBON

### 4.1 OVERVIEW

Figure 1 presents an overview of LLMCarbon for predicting the carbon footprint of an LLM. The inputs to LLMCarbon encompass the LLM's architectural description, data center specification, and hardware configuration. To output the LLM's carbon footprint, LLMCarbon employs a series of models, each processing specific input details. LLMCarbon can use the parameter model to determine the LLM's parameter count based on its architectural attributes, or directly accept the LLM's parameter count as input. With the LLM's parameter count and training token count, LLMCarbon calculates the test loss by the neural scaling law (Kaplan et al., 2020),

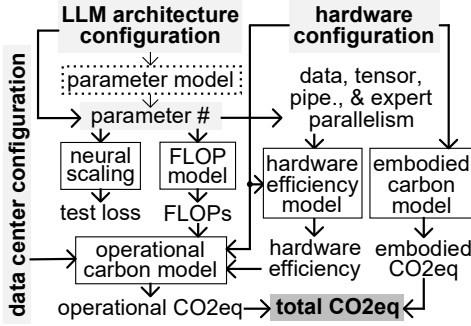

Figure 1: The overview of LLMCarbon.

and employs the FLOP model to estimate the volume of FLOPs required for LLM processing. Through the parameter count, LLMCarbon generates the optimal data, tensor, pipeline, and expert parallelism setting. Taking into account the parallelism setting and hardware configuration, LLMCarbon's hardware efficiency model computes the hardware efficiency, representing the real computing throughput divided by the peak computing throughput. Utilizing data center details, hardware efficiency, and FLOP count, LLMCarbon applies the operational carbon model to derive the LLM's operational carbon footprint. Similarly, by considering the hardware configuration, LLMCarbon's

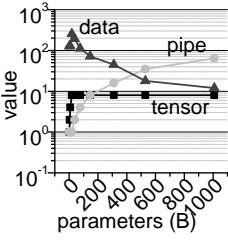 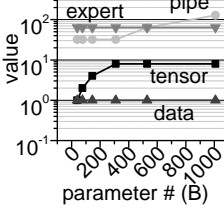 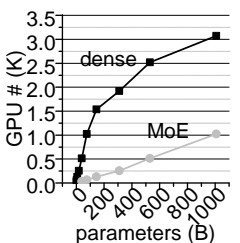 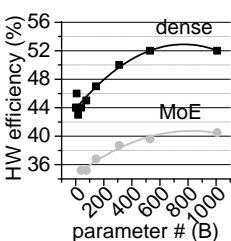

Figure 2: The paral-
lelism setting for pro-
cessing dense LLMs.

Figure 3: The paral-
lelism setting for pro-
cessing MoE LLMs.

Figure 4: The com-
puting device number
for processing LLMs.

Figure 5: The hard-
ware efficiency for
processing LLMs.

embodied carbon model yields the LLM's embodied carbon footprint. The overall carbon footprint of the LLM is then computed by summing both the operational and embodied carbon footprints.

### 4.2 PARAMETER MODEL

Among all LLM architectural attributes, the LLM parameter count has the largest impact on test loss (Kaplan et al., 2020). To reduce projection errors, LLMCarbon can take the parameter count as direct input, or estimate the parameter count by the parameter model. The parameter model's input comprises the LLM's architectural parameters including the hidden size ($h$), the number of layers ($l$), the vocabulary size ($V$), and the number of experts ($N_e$). For a dense LLM, we calculate its parameter count ($P_d$) by Equation 1 (Narayanan et al., 2021). An MoE LLM (Rajbhandari et al., 2022) replaces $\rho$ ($\rho \in (0, 1]$) feed-forward layers in its counterpart dense LLM with MoE layers. An MoE layer's parameter count is the sum of the expert parameter count ($P_{exp} = 8h^2 N_e$) and the self-attention parameter count ($P_{att} = 4h^2$), so the parameter count ($P_e$) of an MoE LLM can be computed using Equation 2. The parameter model of LLMs adopting an encoder-decoder architecture can be viewed in Appendix A.

$$P_d \approx 12lh^2 + Vh \qquad (1) \qquad P_e \approx (1-\rho)P_d + \rho(4h^2 + 8h^2 N_e)l \qquad (2)$$

### 4.3 NEURAL SCALING LAW

The neural scaling law (Kaplan et al., 2020) predicts an LLM's test loss based on its parameter count $P$ and the training dataset size $D$. For ensuring the comparability of test losses across various models, sizes, and datasets, we adopt the Chinchilla scaling law (Hoffmann et al., 2022) formulated as Equation 3, where $A$, $B$, $\alpha$, $\beta$, and $E$ are fitting constants. The test loss $L$ equals to the summation of an irreducible term $E$ and a reducible term diminishing through the scaling of $P$ and $D$.

$$L(P, D) = \frac{A}{P^\alpha} + \frac{B}{D^\beta} + E \quad (3) \qquad TC \approx 6PD \quad (4) \qquad IC \approx 2PD \quad (5)$$

### 4.4 FLOP MODEL

The FLOP model receives two inputs: the count of parameters ($P$) and the number of tokens ($D$) processed by the LLM processing. The primary component of FLOPs is the multiply-accumulate operations involving LLM weights and intermediate results. Within our FLOP model, the FLOP count necessary for training a dense LLM ($TC$) is estimated using Equation 4. For dense LLM inferences, the FLOP count ($IC$) is approximated as per Equation 5. To compute the FLOP count for MoE LLM processing, we input the parameter number of the dense base model (Rajbhandari et al., 2022) of the MoE LLM into Equations 4 and 5, respectively.

### 4.5 HARDWARE EFFICIENCY MODEL

Efficient processing of LLMs relies on achieving high hardware efficiency, which is calculated as the actual computing throughput divided by the peak throughput. This efficiency is largely deter-
mined by the optimal configuration of data, tensor, pipeline, and expert parallelism, along with the number of devices used for the task. Using too few or too many devices or improperly configuring parallelism can lead to reduced hardware efficiency. For example, achieving optimal parallelism for GPT-3 with 175 billion parameters requires 1.5K V100 GPUs, resulting in a hardware efficiency of 47% (Narayanan et al., 2021). Conversely, an unoptimized configuration using 10K V100 GPUs yields a substantially lower hardware efficiency of only 19.7% (Patterson et al., 2021).

**Optimal Parallelism Setting**. The optimal parallelism setting is represented as $(p, t, d, e)$, where each variable corresponds to a degree of pipeline, tensor, data, and expert parallelism, respectively. For dense LLMs, optimal settings are derived from (Narayanan et al., 2021), depicted in Figure 2,

where $e = 1$ is omitted. Initially, we increase tensor parallelism ($t$) up to $z$ (e.g., $z = 8$) when employing $z$-device servers (Narayanan et al., 2021), each containing $z$ interconnected devices. This increment in $t$ is confined to avoid exceeding communication bandwidth limits. Once $z$ is reached, further scaling for larger LLMs involves increasing pipeline parallelism ($p$) (Narayanan et al., 2021). However, the product of $t$ and $p$ ($t \cdot p$) must not exceed a certain threshold to ensure that LLM parameters and intermediate data fit into device memory. The number of devices required to achieve optimal hardware efficiency for dense LLM processing is calculated as $n = t \cdot p \cdot d$ (Narayanan et al., 2021), and can be viewed in Figure 4. A polynomial regression model is used to predict optimal hardware efficiency based on these parameters. For MoE LLMs, the optimal parallelism settings are adopted from (Chen et al., 2023). As Figure 3 shows, assuming 64 experts within an MoE LLM, expert parallelism ($e$) is always set to 64, intertwining $d$ and $e$ for a uniform expert distribution. To reduce inter-device all-to-all communications, $d$ is fixed at 1. Scaling MoE LLM parallelism is achieved by increasing pipeline parallelism ($p$). The number of devices required for optimal hardware efficiency in MoE LLM processing is also calculated as $n = t \cdot p \cdot d$. As Figure 4 exhibits, MoE LLMs require fewer devices compared to dense LLMs with equivalent parameter counts due to their lower computational overhead. The optimal hardware efficiency during MoE LLM processing is represented in Figure 5. MoE LLMs achieve $\sim 80\%$ (Chen et al., 2023) of the optimal hardware efficiency of their dense base models, due to extra host-device memory swaps.

$$eff_{re} = \begin{cases} \gamma_0 \cdot \frac{re}{n} \cdot eff_n & re < n \\ \gamma_1 \cdot \frac{n}{re} \cdot eff_n + \gamma_2 & re > n \end{cases} \quad (6) \qquad t_{dev} = \frac{TFLOP}{n_{dev} \cdot FLOP_{peak} \cdot eff} \quad (7)$$

**Fewer or More Computing Devices**. When the number of computing devices is not equal to $t \cdot p \cdot d$, the hardware efficiency decreases. The efficiency ($eff_{re}$) with $re$ devices can be calculated using Equation 6, where $\gamma_0 \sim \gamma_2$ are fitting constants, $eff_n$ means the highest hardware efficiency, and $n$ indicates the number of devices that can achieve $eff_n$.

$$energy_{hard} = \sum_{i \in hardware\_set} (P_i \cdot eff_i \cdot n_i \cdot t_i) \quad (8) \qquad energy_{oper} = energy_{hard} \cdot PUE \quad (9)$$

## 4.6 Operational Carbon Model

By using the FLOP count ($TFLOP$), the hardware efficiency ($eff$), and the computing device number ($n_{dev}$), we can determine the execution time of a device through Equation 7, where $FLOP_{peak}$ represents the device peak throughput. The total energy ($energy_{hard}$) consumed by all hardware units can be calculated using Equation 8, where $P_i$ denotes the peak power of hardware unit $i$; $eff_i$ represents the hardware efficiency of hardware unit $i$; $n_i$ indicates the count of hardware unit $i$; and $t_i$ means the execution time of hardware unit $i$. Hardware units encompass a range of components, including CPUs, LLM computing devices, memories, SSDs, and others.

$$CO2eq_{oper} = energy_{oper} \cdot carb\_inten \quad (10) \qquad CO2eq_{chip} = area \cdot CPA \quad (11)$$

**PUE**. Power Usage Effectiveness (PUE) (Henderson et al., 2020) serves as the industry standard metric for evaluating a data center's energy efficiency. It is defined as the ratio of the total energy consumption of the data center, including all auxiliary components like cooling, to the energy consumed solely by the computing hardware within the data center. The operational energy ($energy_{oper}$) associated with LLM processing can be calculated using Equation 9, where $energy_{hard}$ denotes the energy used by the computing hardware within a data center, and $PUE$ indicates the PUE of the specific data center.

$$CO2eq_{emb} = \sum_{i \in hardware\_set} \frac{t_i \cdot CO2eq_{chip_i}}{lifetime_i} \quad (12) \qquad CO2eq = CO2eq_{oper} + CO2eq_{emb} \quad (13)$$

**Carbon Intensity**. Carbon intensity is a metric that assesses the environmental impact of a data center's energy consumption. Carbon-free energy (CFE) denotes the proportion of renewable, carbon-free energy utilized within a data center. As a data center increases its utilization of renewable energy, it experiences an increase in CFE and a corresponding decrease in carbon intensity. Table 2 provides insights into the carbon intensity and CFE values for some data centers. The operational carbon footprint ($CO2eq_{oper}$) attributed to LLM processing is calculated using Equation 10, where $energy_{oper}$ represents the operational energy for LLM processing, and $carb\_inten$ denotes the carbon intensity of the specific data center.

Table 2: The data center efficiency.

| data center name | carbon free energy | carbon intensity $gCO2eq/kWh$ |
|---|---|---|
| asia-east2 | 28% | 360 |
| europe-north1 | 91% | 127 |
| us-central1 | 97% | 394 |
| us-south1 | 40% | 296 |

Table 3: The comparison of embodied carbon footprints.

| hardware | description | unit | CPA |
|---|---|---|---|
| CPU | TSMC 16nm | 147 $mm^2$ | 1 $kgCO2/cm^2$ |
| DRAM | Micron 18nm | 256 GB | 0.4 $kgCO2/GB$ |
| SSD | Samsung 20nm | 32 TB | 0.018$kgCO2/GB$ |
| TPUv3 | TSMC 16nm | 700 $mm^2$ | 1 $kgCO2/cm^2$ |
| TPUv4 | TSMC 7nm | 400 $mm^2$ | 1.6 $kgCO2/cm^2$ |
| V100 | TSMC 12nm | 815 $mm^2$ | 1.2 $kgCO2/cm^2$ |
| H100 | TSMC 4nm | 814 $mm^2$ | 1.8 $kgCO2/cm^2$ |

### 4.7 EMBODIED CARBON MODEL

To quantify the chip's embodied carbon footprint ($CO2eq_{chip}$) within a specific hardware unit, Equation 11 is employed, where $area$ represents the chip's area. The Carbon emitted Per unit Area ($CPA$) is contingent on various semiconductor fabrication parameters, including yield, energy consumption per unit area during manufacturing, emissions from chemicals utilized in hardware production, and emissions associated with raw material sourcing for fabrication. Specific values for area and CPA for distinct hardware units are elaborated in Table 3, where area values for CPU, DRAM, SSD, TPU, and GPU are drawn from sources such as (Singh et al., 2020), (Choe, 2021), (Wiki, 2023b), and (Wiki, 2023a). CPA values for Micron, Samsung, and TSMC are extracted from (Garcia Bardon et al., 2020), and (TSMC, 2019). The total embodied carbon footprint ($CO2eq_{emb}$) originating from all hardware units involved in LLM processing is assessed using Equation 12, where $CO2eq_{chip_i}$ denotes the chip's embodied carbon footprint for hardware unit $i$, $lifetime_i$ means the lifespan of hardware unit $i$, and $t_i$ represents the execution duration of hardware unit $i$. The hardware units mentioned in Equation 12 include CPUs, LLM computing devices, memories, SSDs, and other components. Notably, Meta's data centers achieve an average utilization rate of $60\%$ throughout the 5-year lifespan of hardware units (Wu et al., 2022).

### 4.8 TOTAL CARBON FOOTPRINT

The total carbon footprint ($CO2eq$) resulting from LLM processing is determined using Equation 13, where $CO2eq_{oper}$ indicates the operational carbon footprint of the LLM, and $CO2eq_{emb}$ denotes the embodied carbon footprint of the LLM.

## 5 VALIDATION

We employ LLMCarbon to compute the operational footprints of five LLMs, including dense and MoE architectures, developed by Google, OpenAI, and Meta during their training phases. We also compute the operational footprint of another LLM, Noor (Lakim et al., 2022), during its storage phase. To validate the predictions of LLMCarbon, we compare our calculated operational footprint values with the previously published data for these LLMs. Moreover, we utilize LLMCarbon to predict the embodied footprint of an LLM developed by Meta and validate the result by comparing it with the actual embodied footprint data.

### 5.1 OPERATIONAL CARBON FOOTPRINT VALIDATION

**Training Phase**. Table 4 presents the validation results of LLMCarbon's predictions on the training operational carbon footprint. To validate the training operational carbon footprint estimations yielded by LLMCarbon, we selected five LLMs: T5 (Raffel et al., 2020), GPT-3 (Brown et al., 2020), GShard (Lepikhin et al., 2021), Switch (Fedus et al., 2022), and XLM (Conneau et al., 2020). We list the inputs and outputs of LLMCarbon in Table 4. Within the table, "device TPD (W)" indicates the Chip Thermal Design Power of a computing device, while "avg. system power (W)" conveys the average system power per computing device, including TPU/GPU, host CPU, DRAM, and network interface. The inputs on the parameters of LLMs, hardware, and data centers, and the actual training operational carbon footprint values of these LLMs were collected from (Patterson et al., 2021) and (Wu et al., 2022). Since the parameter count of an LLM is considered as an architectural parameter of the LLM in (Patterson et al., 2021) and (Wu et al., 2022), we skipped the parameter model and directly used the parameter count as an input to LLMCarbon. The validation of the parameter

Table 4: The validation on the operational carbon footprints of various LLMs.

| LLM | T5 | GPT3 | GShard | Switch | XLM |
|---|---|---|---|---|---|
| reference | (Patterson et al., 2021) | | | | (Wu et al., 2022) |
| developer | Google | OpenAI | Google | Google | Meta |
| type | dense | dense | MoE | MoE | dense |
| parameter # (B) | 11 | 175 | 619 | 1500 | 0.55 |
| base model param. # (B) | - | - | 2.3 | 7.41 | - |
| token # (B) | 500 | 300 | 1K | 2K | 7K |
| $CO_2 eq/KWh$ | 0.545 | 0.429 | 0.177 | 0.33 | 0.413 |
| PUE | 1.12 | 1.1 | 1.09 | 1.1 | 1.1 |
| computing device | TPUv3 | V100 | TPUv3 | TPUv3 | V100 |
| device TPD (W) | 450 | 300 | 450 | 450 | 300 |
| avg. system power (W) | 310 | 330 | 288 | 245 | 342 |
| peak TFLOPs/s | 123 | 125 | 123 | 123 | 125 |
| achieved TFLOPs/s | 45.6 | 24.6 | 48 | 34.4 | 26.5 |
| hardware efficiency | 37% | 19.7% | 39% | 28% | 21.2% |
| device # | 512 | 10K | 1K | 1K | 512 |
| total zettaFLOPs | 40.5 | 314 | 13.3 | 82.2 | 23.9 |
| training days | 20 | 14.8 | 3.1 | 27 | 20.4 |
| actual $tCO_2 eq$ | 46.7 | 552.1 | 4.3 | 59.1 | 39 |
| mlco2 predicted $tCO_2 eq$ | 89.4 | 955.2 | 8.4 | 137.3 | 66.96 |
| mlco2 $\Delta$ | +91.3% | +73% | +95.3% | +132% | +69% |
| **LLMCarbon predicted** $tCO_\mathbf{2} eq$ | 45.66 | 553.87 | 4.46 | 63.9 | 37.6 |
| **LLMCarbon $\Delta$** | $-\mathbf{2.22\%}$ | $+\mathbf{0.32\%}$ | $+\mathbf{3.8\%}$ | $+\mathbf{8.2\%}$ | $-\mathbf{3.54\%}$ |

model of LLMCarbon can be found in Appendix B. Owing to the adoption of suboptimal parallelism settings, the hardware efficiencies for training these LLMs hover within the range of 39% to 19.7%, lower than the hardware efficiencies achieved with optimal parallelism configurations. Comparing the predicted operational carbon footprints to actual data, LLMCarbon's projections display disparities of $\leq 8.2\%$. When predicting the operational carbon footprint during the training of MoE LLMs, LLMCarbon incurs a higher margin of error, due to the intricacy of MoE architectures. On the contrary, when compared to actual data, the training operational carbon footprint estimations made by mlco2 (Lacoste et al., 2019) suffer from huge disparities of more than 69%, because mlco2 assumes all devices consistently operate at the peak computing throughput and consume the peak power.

**Inference Phase**. To validate the operational carbon footprint predictions generated by LLMCarbon, we consider the inferences of GPT3 with 175B parameters (Yu et al., 2022). These inferences were carried out on 16 A100 GPUs, using a batch size of 32 and an input size of 128 tokens (Yu et al., 2022). According to the hardware efficiency model, this specific hardware configuration yields a hardware efficiency of 9.26%. Achieving the optimal hardware efficiency for GPT3 requires ∼1.5K GPUs, which is significantly more than what was used for these inferences. LLMCarbon's predicted latency for this inference batch is 3.1s, while the actual latency for this inference batch is 3s (Yu et al., 2022). We assume the inference experiments took place in a data center with a PUE of 1.1 and a carbon intensity of 0.429 $CO_2 eq/KWh$. The difference between the predicted and actual inference operational carbon footprints does not exceed +3.3%.

**Storage Phase**. The typical power consumption of cloud storage is reported as 11.3W/TB (Posani et al., 2018), while the power consumption for data transfer within a data center is around 1.48W/TB (Baliga et al., 2011). Over a six-month storage phase, the Noor LLM (Lakim et al., 2022) encompasses 32.7TB of storage data, comprising curated data, bulk data, and the model. Additionally, it transfers a data volume of 277.4TB. Based on LLMCarbon's estimations, the storage data energy is predicted as 1.596MWh (compared to the actual 1.69MWh (Lakim et al., 2022)), while the energy consumption attributed to data transfer is projected to be 1.77MWh (compared to 1.8MWh (Lakim et al., 2022)). Notably, the projection accuracy of LLMCarbon regarding the operational energy during the storage phase showcases an error margin of less than 3.6%.

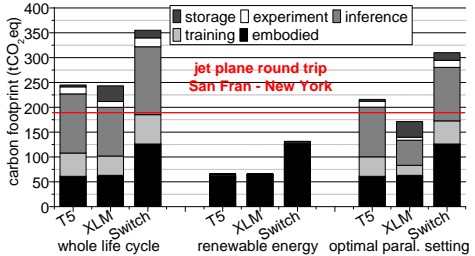

Figure 6: The carbon footprint of three LLMs in case studies.

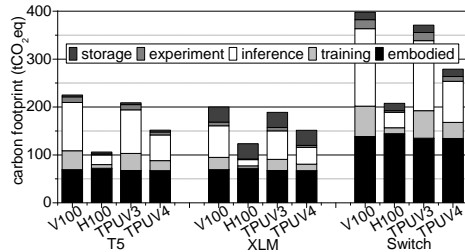

Figure 7: The carbon footprint of GPT3 trained by different computing devices.

**Experimentation Phase**. The experimentation phase consisting of various activities of training, inference, and storage (Wu et al., 2022). And we have validated the training phase, inference phase, and storage phase of an LLM in previous sections.

## 5.2 EMBODIED CARBON FOOTPRINT VALIDATION

Table 5 presents the validation results of the embodied carbon footprint estimated by LLMCarbon in comparison to the published data of XLM (Wu et al., 2022). This is the only publicly available data regarding the embodied carbon footprint of a LLM training hardware infrastructure to our best knowledge. The setup consists of 512 V100 GPUs organized into 64 8-GPU servers, each equipped with a CPU, a 32TB SSD disk, and a 256GB DRAM main memory system. Using the unit and CPA data from Table 3, we computed the

Table 5: The embodied carbon footprint validation against Meta XLM.

| hardware | number | $CO2eq_{chip}$ $(kgCO_2eq)$ | $\frac{time}{lifetime}$ | $CO2eq_{emb}$ $(tCO_2eq)$ |
|---|---|---|---|---|
| GPU | 512 | 9.78 | 1.12% | 0.056 |
| CPU | 64 | 1.47 | 1.12% | 0.0018 |
| SSD | 64 | 576 | 1.12% | 0.412 |
| DRAM | 64 | 102.4 | 1.12% | 0.073 |
| others | 64 | 148.2 | 1.12% | 0.096 |
| **predicted sum** | | | | 0.64 |
| actual 0.66 $tCO_2eq$, $\Delta$ $-3.05\%$ | | | | |

values of $CO2eq_{chip}$ presented in Table 5. The training duration of XLM is 20.4 days, and Wu et al. (2022) assumed a hardware unit lifetime of 5 years. Consequently, the $\frac{time}{lifetime}$ values for all hardware units were determined to be 1.12%. Apart from CPU, GPU, SSD, and DRAM, other hardware components (others) such as the motherboard, chassis, and PSU collectively contribute to 15% (Tannu & Nair, 2022) of the anticipated total embodied carbon footprint. In contrast to the reported embodied carbon footprint of XLM (Wu et al., 2022), the predictions produced by LLM-Carbon reveal a disparity of $-3.05\%$.

## 6 CASE STUDIES USING LLMCARBON

We used LLMCarbon to demonstrate the following case studies.

**Large Embodied Carbon Footprint**. The embodied carbon footprint throughout the life-cycle of an LLM is significant. Even when no computing activities occur, the LLM still incurs embodied carbon overhead due to the idle hardware allocated to the LLM. As illustrated in Figure 6, the embodied carbon footprint of an LLM across its entire life-cycle contributes to approximately $24\% \sim 35\%$ of the overall carbon footprint (including embodied, training, inference, experimentation, and storage carbon footprints) of the LLM. We adopted the ratio between training, inference, and experimentation activities from (Wu et al., 2022). Furthermore, as data centers progressively shift towards adopting renewable energy sources, the embodied carbon footprint of an LLM will dominate the entire life-cycle carbon footprint of the LLM in the near future. For instance, 97% of the operational energy in a Meta data center (Wu et al., 2022) is provided by renewable sources. The embodied carbon footprints of diverse LLMs operating within this data center constitute $92\% \sim 95\%$ of their entire life-cycle carbon footprints. This underscores the pivotal role of accounting for embodied carbon in the sustainability evaluation of LLMs.

**Optimal Parallelism Setting**. As discussed in Section 5.1, the training processes of the LLMs used in our validation lacked optimized parallelism settings. By using LLMCarbon, we pinpoint the optimal configurations for data, tensor, pipeline, and expert parallelism pertaining to these three

LLMs. As illustrated in Figure 6, the adoption of these optimal parallelism settings leads to a noteworthy decrease (i.e., $16\% \sim 39\%$) in their operational carbon footprints.

**New Accelerators**. When employing distinctive computing devices for the LLM processing, the operational carbon footprints of an LLM tend to differ, while the embodied carbon footprints remain similar. Figure 7 showcases the outcomes derived from training, inferring, and experimenting with three LLMs utilizing V100 GPU, H100 GPU, TPUv3, and TPUv4. Their embodied carbon footprints exhibit similarity, as the embodied carbon emissions of SSD and DRAM dominate their total embodied carbon footprints. However, compared to V100 GPUs, the operational carbon footprints of these LLMs are notably curtailed by 71% and 41% when employing H100 and TPUv4 accelerators, respectively. Embracing novel computing devices for LLMs presents a pragmatic path to mitigate their operational carbon footprints.

**Training Carbon Footprint Scaling**. In addition to the LLMs (i.e., T5, GPT3, GShard, Switch, XLM, and Noor) we used in validations, we also included other LLMs in our analysis, such as PaLM (Chowdhery et al., 2022), Gopher (Rae et al., 2021), Chinchilla (Hoffmann et al., 2022), LaMDA (Thoppilan et al., 2022), Jurassic-1 (Lieber et al., 2021), MT-NLG (Smith et al., 2022), Bloom (Scao et al., 2022), YaLM (Yandex, 2022), GLM (Zeng et al., 2023), GLaM (Du et al., 2022), FB-MoE (Artetxe et al., 2021), ST-MoE (Zoph et al., 2022), and PR-MoE (Rajbhandari et al., 2022). Among these LLMs, GShard, Switch, GLaM, FB-MoE, ST-MoE, and PR-MoE use sparse MoE architectures, while the other LLMs adopt dense architectures. We do not aim to directly compare the accuracy and carbon emissions of these original LLMs, since they were trained by different datasets and in different data centers. Instead, we study the test losses and training operational carbon footprints

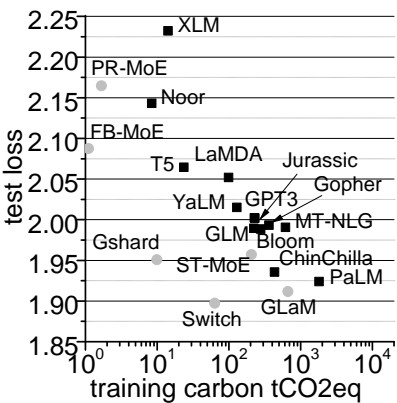

Figure 8: The trade-off between training carbon footprint and test loss.

of some new LLM designs adopting the same architectures as these LLMs. We assume these new LLM designs are trained using the same dataset and the same hardware infrastructure in the same data center. We present the test losses and training operational carbon footprints of these LLMs in Figure 8. To compute the test loss, we adopt the fitting constants including $\alpha = 0.34$, $\beta = 0.28$, $A = 406.4$, $B = 410.7$, and $E = 1.69$ for Equation 3 from (Hoffmann et al., 2022). Since the test loss of an MoE LLM with $P$ parameters is similar to that of its dense counterpart with only $P/8$ parameters (Rajbhandari et al., 2022), we decreased the $P$ of MoE LLMs to $P/8$ in Equation 3. The training processes of all LLMs use their optimal parallelism settings and the corresponding numbers of V100 GPUs hosted by a data center where PUE is 1.1 and $CO_2 eq/KWh$ is 0.431. Overall, an LLM with a larger number of parameters and trained on more tokens achieves a lower test loss but also consumes a larger training operational carbon footprint. Compared to dense LLMs, the Pareto front of MoE LLMs is closer to the origin point, indicating that an MoE LLM can obtain a lower test loss by the same training carbon footprint.

## 7 CONCLUSION

In this paper, we propose LLMCarbon, an end-to-end carbon footprint modeling tool for dense and MoE LLMs, which contribute significantly to carbon emissions during training, inference, experimentation, and storage processes. LLMCarbon can accurately assess the operational and embodied carbon footprints of an LLM, enabling efficient exploration of the design space by considering the trade-off between carbon footprint and test loss. It also promotes the adoption of carbon footprint reduction measures by facilitating quantitative comparisons among various LLM configurations.

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

Table 6: The architectural details of dense LLMs for validations and explorations. The dense LLMs we selected include T5 (Raffel et al., 2020), GPT-3 (Brown et al., 2020), XLM (Conneau et al., 2020), Noor (Lakim et al., 2022), PaLM (Chowdhery et al., 2022), Gopher (Rae et al., 2021), Chinchilla (Hoffmann et al., 2022), LaMDA (Thoppilan et al., 2022), Jurassic-1 (Lieber et al., 2021), MT-NLG (Smith et al., 2022), Bloom (Scao et al., 2022), YaLM (Yandex, 2022), and GLM (Zeng et al., 2023).

| Name | Param.(B) | $V$ | $h$ | $d_{ff}$ | $d_{head}$ | $N_{head}$ | $l$ | Equ. | $P_d$ (B) | Diff.$\Delta$ |
|------|-----------|-----|-----|----------|------------|------------|-----|------|-----------|---------------|
| T5 | 11 | 32K | 1024 | 65536 | 128 | 128 | 24 | 14 | 11.3 | +2.79% |
| GPT3 | 175 | 51.2K | 12288 | 49152 | 128 | 96 | 96 | 1 | 174.58 | -0.24% |
| XLM | 0.55 | 250K | 1024 | 4096 | 64 | 16 | 24 | 1 | 0.557 | +1.45% |
| Noor | 13 | - | - | - | - | - | - | - | - | - |
| PaLM | 540 | 256K | 18432 | 73728 | 256 | 48 | 118 | 15 | 539.24 | -0.14% |
| Gopher | 280 | 51.2K | 16384 | 65536 | 128 | 128 | 80 | 1 | 258.54 | -7.66% |
| Chinchilla | 70 | 51.2K | 8192 | 32768 | 128 | 64 | 80 | 1 | 64.84 | -7.36% |
| LaMDA | 137 | 51.2K | 8192 | 65536 | 128 | 128 | 64 | 15 | 137.86 | +0.63% |
| Jurassic-1 | 178 | 256K | 13824 | 55296 | 144 | 96 | 76 | 1 | 175 | -1.68% |
| MT-NLG | 530 | 51.2K | 20480 | 81920 | 160 | 128 | 105 | 1 | 529.53 | -0.09% |
| Bloom | 176 | 51.2K | 14336 | 57344 | 128 | 112 | 70 | 1 | 173.37 | -1.49% |
| YaLM | 100 | - | - | - | - | - | - | - | - | - |
| GLM | 130 | 51.2K | 12288 | 49152 | 128 | 96 | 70 | 1 | 127.46 | -1.95% |

## A   MORE ON THE LLM PARAMETER MODEL

We listed the architectural parameters of dense LLMs we selected in Table 6, and the architectural parameters of MoE LLMs we used in Table 7.

**GPT3-like Dense LLMs**: The parameter count for most dense LLMs structured on a GPT3-like architecture (Brown et al., 2020) can be determined using Equation 1. In each layer of these dense LLMs, there exists a self-attention layer and a feed-forward layer. The $W_q$, $W_k$, and $W_v$ matrices of the self-attention layer possess dimensions of $hN_{head}d_{head}$, with $h$ representing the hidden size, $N_{head}$ indicating the number of heads, and $d_{head}$ denoting the head dimension. The $W_o$ matrix that links the self-attention layer to the feed-forward layer also has a dimension of $hN_{head}d_{head}$. In the feed-forward layer, two $hd_{ff}$ weight matrices are used, where $d_{ff}$ signifies the dimension of the feed-forward layer. In a conventional LLM architecture, we have $n_{head}d_{head} = h$ and $d_{ff} = 4h$. Consequently, the parameter count for a single dense LLM layer can be calculated as $4hN_{head}d_{head} + 2hd_{ff} = 12h^2$. Additionally, a dense LLM possesses $Vh$ token embedding parameters, where $V$ denotes the vocabulary size. In total, a dense LLM utilizing a GPT3-like architecture incorporates $12h^2l + Vh$ parameters, where $l$ stands for the number of layers.

**Encoder-Decoder Dense LLMs**: Certain dense LLMs from Google, such as T5 (Raffel et al., 2020), employ an encoder-decoder transformer architecture. Within a single layer of these LLMs, there exist both an encoder and a decoder. The encoder comprises a self-attention layer and a feed-forward layer, while the decoder includes two self-attention layers and a feed-forward layer. The parameter count for the encoder is $4hN_{head}d_{head} + 2hd_{ff}$, whereas the parameter count for the decoder is $8hN_{head}d_{head} + 2hd_{ff}$. Therefore, the total parameter count for a single LLM resembling T5 becomes $12hN_{head}d_{head} + 4hd_{ff}$. In the case of a T5-like LLM, where $n_{head}d_{head} \neq h$ and $d_{ff} \neq 4h$, we cannot derive a further simplified equation. The overall parameter count for a T5-like LLM can be estimated as:

$$P_d \approx (12hN_{head}d_{head} + 4hd_{ff})l + Vh. \tag{14}$$

For some dense LLMs like LaMDA (Thoppilan et al., 2022), which consist of only a decoder in each layer, the total parameter count for a LaMDA-like LLM is:

$$P_d \approx (8hN_{head}d_{head} + 2hd_{ff})l + Vh. \tag{15}$$

**MoE LLMs**: In the case of certain MoE LLMs, especially those developed by Google, we also encounter scenarios where $n_{head}d_{head} \neq h$ and $d_{ff} \neq 4h$. Consequently, within an MoE layer,

Table 7: The architectural details of MoE LLMs for validations and explorations. The MoE LLMs we selected include Gshard (Lepikhin et al., 2021), Switch (Fedus et al., 2022), GLaM (Du et al., 2022), FB-MoE (Artetxe et al., 2021), ST-MoE (Zoph et al., 2022), and PR-MoE (Rajbhandari et al., 2022).

| Name | Param.(B) | $P_d$ (B) | $\rho$ | $h$ | $d_{ff}$ | $d_{head}$ | $N_{head}$ | $l$ | $N_e$ | Equ. | $P_e$ (B) | Diff.$\Delta$ |
|------|-----------|-----------|--------|-----|----------|------------|------------|-----|-------|------|-----------|---------------|
| Gshard | 600 | 2.3 | 0.5 | 1024 | 8192 | 128 | 16 | 36 | 2048 | 16 | 618.47 | +3.07% |
| Switch | 1571 | 7 | 1 | 2048 | 6144 | 32 | 64 | 15 | 2048 | 16 | 1546.19 | -1.58% |
| GLaM | 1200 | 95 | 0.5 | 8192 | 32768 | 128 | 128 | 64 | 64 | 2 | 1133.87 | -5.51% |
| FB-MoE | 1100 | 2.3 | 0.5 | 4096 | 16384 | 128 | 32 | 32 | 512 | 2 | 1103.81 | +0.35% |
| ST-MoE | 269 | 32 | 0.25 | 5120 | 20480 | 128 | 64 | 27 | 64 | 16 | 273.17 | +1.55% |
| PR-MoE | 31 | 1.3 | 0.5 | 2048 | 8192 | 128 | 16 | 24 | 64/128 | 16 | 31.8 | +2.5% |

we can compute the expert parameter count as $P_{exp} = 2hd_{ff}N_e$, while the self-attention parameter count can be determined as $P_{att} = 4hn_{head}d_{head}$. The overall parameter count for such an MoE LLM can be estimated as follows:

$$P_e \approx (1 - \rho)P_d + \rho(2hd_{ff}N_e + 4hn_{head}d_{head})l \tag{16}$$

# B PARAMETER MODEL VALIDATION

**Validation of Dense LLMs**: We present the architectural parameters of dense LLMs in Table 6. It's worth noting that while Noor was utilized in the validation of training operational energy and YaLM in LLM scaling, their original papers (Lakim et al., 2022; Yandex, 2022) do not provide architectural specifications, thus preventing us from determining their parameter count using LLMCarbon. In Table 6, we apply Equation 1 to calculate the parameter count for models such as GPT3, XLM, Gopher, Chinachilla, Jurassic-1, MT-NLG, Bloom, and GLM. Additionally, we use Equation 14 to estimate the parameter count for T5, and Equation 15 for PaLM and LaMDA. Among all dense LLMs, Gopher and Chinchilla exhibit the most substantial disparities between the predicted and actual parameter numbers. This deviation is primarily attributed to the usage of positional encoding mechanisms in these LLMs, with the weights employed in their relative positional encodings not included in our calculations. For instance, Gopher incorporates 21.5 billion weights in its relative positional encoding, contributing to this observed difference.

**Validation of MoE LLMs**: We present the architectural parameters of MoE LLMs in Table 7. To compute the parameter count for GLaM, and FB-MoE, we utilize Equation 2. For Gshard, Switch, ST-MoE, and PR-MoE, we apply Equation 16. In PR-MoE, a portion of MoE layers have 64 experts, and the other MoE layers have 128 experts. In the table, Gshard, GLaM, and PR-MoE encounter the largest disparities between the predicted and actual parameter counts. The deviation is caused by the usage of positional encoding mechanisms in these MoE LLMs, and the unaccounted parameters in their routing networks.

