# OpenReview forum: "LLMCarbon: Modeling the End-to-End Carbon Footprint of Large Language Models"
_ICLR.cc/2024/Conference — ICLR 2024 oral_

### Official Review · Reviewer_12is · 2023-10-31

**Soundness:** 4 excellent
**Presentation:** 4 excellent
**Contribution:** 4 excellent
**Rating:** 10
**Confidence:** 4

**Summary:**

The work builds an analytic model that predicts carbon equivalent emissions of LLM training and deployment. The model is extensive and contains variables and functions such as scaling laws, parallelism used, mixture of experts, data center efficiencies, and embodied carbon footprint.

Recommendation: While the model could further be extended to include more detailed variables, it is a groundbreaking effort that will lay the foundation for all future carbon models. I recommend this work to be highlighted at the conference.

**Strengths:**

- a very comprehensive model that captures previously neglected variables that led to widely inaccurate carbon emission predictions
- uses scaling laws and other variables to make accurate predictions of the carbon emissions of any training run
- validation of the model with reported carbon footprints in the literature.

**Weaknesses:**

- the mixture of expert models seems to use regular scaling laws equations for performance prediction. More appropriate would be to use the equations from the routed scaling laws paper.
- some factors are not very well discussed. For example, experimentation can have a large variance of CO2eq used. For deployment, more experimentation is usually undertaken to improve model efficiency through speculative decoding and distillation. A short discussion on the most CO2eq intensive factors and how they might differ between companies/institutions would be appropriate (no need to model this)

**Questions:**

Comments:
- Figure 6 and 7 caption seem to be swapped

Questions:
- why do you use 16 A100s, batch size 32 and 128 tokens for inference benchmarking? I would assume the most common deployment strategies for inference are (1) 8x A100 + NVSwitch with batch size of 64-128 for token-by-token generation and (2) personal deployment with batch size 1 (this can be significant for T5 and other open source LLMs which enjoy large widespreadnn use)

---

> ### Author Response · Authors · 2023-11-15
>
> **Thank Reviewer 12is for the careful and insightful review of our manuscript!**
>
> ## 1. The mixture of expert models seems to use regular scaling laws equations for performance prediction. More appropriate would be to use the equations from the routed scaling laws paper.
>
> The reviewer's observation regarding the scaling behavior of MoE LLMs is indeed valid. We have noted that MoE LLMs exhibit scaling patterns distinct from those of dense LLMs. However, in the context of the "Training Carbon Footprint Scaling" subsection within Section 6, when comparing MoE and dense LLMs, we have employed the same scaling law for both MoE and dense LLMs. Considering that the test loss of an MoE LLM with P parameters closely resembles that of its dense counterpart with only P/8 parameters, as demonstrated by Rajbhandari et al. (2022), we have opted to adjust the value of P for MoE LLMs to P/8 within Equation 3. We acknowledge the need to refine this approach and employ routed scaling laws to enhance the accuracy of modeling MoE LLMs in future iterations of our work.
>
>
> ## 2. Some factors are not very well discussed. For example, experimentation can have a large variance of CO2eq used. For deployment, more experimentation is usually undertaken to improve model efficiency through speculative decoding and distillation. A short discussion on the most CO2eq intensive factors and how they might differ between companies/institutions would be appropriate (no need to model this).
>
> Our primary aim is to establish a baseline framework with the capacity to delineate the carbon footprint attributed to LLMs. It is essential to underscore that the current investigation does not encompass certain intricate algorithmic optimizations, including but not limited to speculative decoding, model distillation, and token pruning. Furthermore, we have refrained from incorporating advanced hardware enhancements, such as aggressive quantization techniques (utilizing formats such as fp16, int8, and int4). Our intention is to delve into the integration of these advanced features in our forthcoming research endeavors. We will add a short discussion on the most CO2eq intensive factors and how they might differ between institutions in the next version of this manuscript.
>
>
> ## 3. Why do you use 16 A100s, batch size 32 and 128 tokens for inference benchmarking? I would assume the most common deployment strategies for inference are (1) 8x A100 + NVSwitch with batch size of 64-128 for token-by-token generation and (2) personal deployment with batch size 1 (this can be significant for T5 and other open source LLMswhich enjoy large widespreadnn use)
>
> The rationale behind our utilization of 16 A100 GPUs, a batch size of 32, and a token count of 128 for conducting inferences resides in the imperative requirement to validate the LLM inference phase using LLMCarbon. The validation data is sourced from (Yu et al., 2022), where these specific configurations—namely, 16 A100 GPUs, batch size 32, and 128 tokens—were employed. It is paramount for us to replicate these configurations in their entirety to ensure the fidelity of the inference carbon footprint generated for the purpose of validating LLMCarbon.

---

### Official Review · Reviewer_pKfr · 2023-10-31

**Soundness:** 2 fair
**Presentation:** 3 good
**Contribution:** 2 fair
**Rating:** 5
**Confidence:** 4

**Summary:**

The main contribution of this paper is to present a comprehensive model for end-to-end carbon footprint of LLMs. Observing that LLMs can be a major contributed to overall carbon footprint, the authors model its carbon footprint from both operational and embodied perspectives, and presents detailed results for different types of architectures.

**Strengths:**

Modeling carbon footprint is an interesting (and increasingly important) research direction. This submission is probably one of the early papers on the topic.

**Weaknesses:**

Not clear whether the authors' definition of embodied carbon is accurate.
Not clear where the carbon coefficient factors in in the model formulation.

**Questions:**

While the paper certainly addresses an important topic, there are at least three issues with it, from the perspective of this reviewer:


1) It seems to me that the authors definition of embodied carbon is the carbon emitted during the production of the hardware on which LLM models run. If that is the case, why? Because that hardware is not just used for running LLM workloads; it is. probably used for running other workloads as well. In my humble opinion, the embodied carbon -- in the case of LLMs -- should represent the carbon emitted during the training, as trained model is the main product/output of LLM. What do authors think about this?

2) It is not immediately evident to me how carbon (emission) factor is accounted for in this modeling. In principle, this coefficient is location dependent and consequently depending on where you are ruining your LLMs, you should get a different carbon footprint. Can the authors clarify that?

3) While I appreciate the importance of carbon modeling of LLMs, it is not clear from this work i) whether the paper is relevant to ICLR as it does not address any core topic in learning, and ii) how it can be used for optimization purposes.

---

> ### Author Response · Authors · 2023-11-15
>
> **Thank Reviewer pKfr for the careful and insightful review of our manuscript!**
>
> ## 1. Not clear where the carbon coefficient factors in the model formulation. It seems to me that the author’s definition of embodied carbon is the carbon emitted during the production of the hardware on which LLM models run. If that is the case, why? Because that hardware is not just used for running LLM workloads; it is. probably used for running other workloads as well.
>
> We have adopted the definition of the embodied carbon footprint for machine learning models as delineated (Wu et al., 2022) and (Tannu & Nair, 2022). The embodied carbon footprint quantifies the quantity of carbon emissions generated during the manufacturing of the hardware, subsequently multiplied by the processing duration of an LLM. This resultant value is then divided by the overall lifespan of the hardware. It is important to acknowledge that the hardware infrastructure within data centers serves not only the processing needs of LLMs but also supports various other computational workloads. This consideration has been duly taken into account in our analysis. In Table 5, it is evident that the training latency associated with Meta XLM accounts for a mere 1.12% of the total projected lifespan of the hardware upon which it operates. We have made the assumption that the hardware is designed to endure for a period of 5 years. Furthermore, our Equation 12 underscores the intrinsic relationship between the processing duration of an LLM and the anticipated lifespan of the hardware hosting the LLM.
>
> ## 2. It is not immediately evident to me how carbon (emission) factor is accounted for in this modeling. In principle, this coefficient is location dependent and consequently depending on where you are ruining your LLMs, you should get a different carbon footprint. Can the authors clarify that?
>
> The reviewer's observation regarding the dependency of the coefficient on geographical location, and consequently, on the location where LLMs are operated, is indeed accurate. The influence of the data center's location is explicitly accounted for through the incorporation of Power Usage Effectiveness (PUE) and carbon intensity in Equation 9 and Equation 10. It is worth noting that distinct data centers exhibit varying values for both Power Usage Effectiveness (PUE), as expounded upon by Henderson et al. (2020), and carbon intensity. PUE is a metric defined as the ratio of the overall energy consumption of a data center, encompassing all ancillary components such as cooling, to the energy consumed exclusively by the computing hardware housed within the data center. On the other hand, carbon intensity serves as a metric that evaluates the environmental impact stemming from the energy consumption of a data center. In Table 4, one can discern the disparities in CO2eq/KWh and PUE values across different data centers, underscoring the significance of geographical location in determining these metrics.
>
> ## 3. While I appreciate the importance of carbon modeling of LLMs, it is not clear from this work i) whether the paper is relevant to ICLR as it does not address any core topic in learning, and ii) how it can be used for optimization purposes.
>
> We sincerely appreciate the reviewer's acknowledgment of the significant role that sustainability plays in the development LLMs. Our LLMCarbon tool represents a pioneering effort in accurately modeling both the operational and embodied carbon footprints of LLMs, a facet of research that undeniably aligns with the core themes of societal considerations at ICLR.
>
> In Section 6 of our manuscript, we have meticulously presented an extensive analysis comprising four distinct user case studies, each serving to elucidate the valuable applications of our LLMCarbon tool. These case studies serve as exemplary instances of how LLMCarbon can be effectively employed for optimization purposes. The four case studies are as follows:
> 1. Impact of Renewable Energy Adoption in Data Centers: Within this context, we thoroughly investigate the impact of incorporating renewable energy sources into data centers, specifically in relation to their influence on the carbon footprint of LLMs.
> 2. Optimal Parallelism Settings in LLM Training: This case study offers an in-depth exploration of the consequences associated with the adoption of optimal parallelism settings during the training of LLMs, with a particular focus on their carbon footprint implications.
> 3. Adoption of New Hardware Accelerators (TPUv3 and TPUv4): Our investigation delves into the implications and advantages of integrating advanced hardware accelerators, namely TPUv3 and TPUv4, across all phases of LLM processing.
> 4. Training Carbon Footprint Scaling Across Various LLMs: This case study delves into the concept of Training Carbon Footprint Scaling, providing insights into the intricate interplay between the carbon footprint, test loss, and the architectural parameter count of different LLMs.

---

> > ### Comment · Reviewer_pKfr · 2023-11-22
> >
> > Thank you for answering my concerns. Although I am still not convinced about the embodied carbon definition adopted in this work, I will raise my score (as the last two concerns of mine have been addressed well).

---

### Official Review · Reviewer_5gUj · 2023-10-31

**Soundness:** 3 good
**Presentation:** 2 fair
**Contribution:** 3 good
**Rating:** 6
**Confidence:** 4

**Summary:**

The paper provides a tool for estimating the carbon footprint of large language models (LLMs). The tool accounts for each life-cycle phase of an LLM: embodied carbon, training, inference, storage. The energy used by hardware is accounted for by considering data center characteristics and model architecture attributes. Comparison with published carbon numbers shows less than 10% error in estimation.

**Strengths:**

- Problem statement is clear
- Carbon footprint considers a comprehensive view of LLM life-cycle
- Use of real-world data points from literature to perform carbon footprints
- Validation against published carbon footprint numbers
- Low error rates in estimation

**Weaknesses:**

- It is not clear how the "ground truth" carbon footprint numbers were established by prior works. Did they perform actual measurements? This is a concern because if the numbers are estimated, and there is an overlap in estimation methods, then it's not surprising that the error rate is low.
- It is difficult to separate the contributions of the paper from the prior works. Much of the equations and parameters of the equations are based on prior work. Is the contribution of the paper to sum up carbon numbers from prior work equations?
- There is no measure of utility of an LLM. For example, one can reduce the emissions by increasing test loss, but a high test loss can lead to poor performance which cannot be used in the real-world. It'll be interesting to measure the utility across classical ML models such as those used for classification, regression, translation, etc.
- It would be good to discuss the utility of the measurement tool. Is the expectation that model designers pick an appropriate data center, architecture, or training dataset to use based on carbon footprint? Much of the time such decisions are constrained by other requirements. For example, model designers have little control over the embodied footprint of the ML model.

**Questions:**

- Are ground truth carbon footprint values based on actual measurements?
- Did you tune your equations or its parameters to reduce the error? If so, how do we know that LLMCarbon is not "overfitting"?

---

> ### Author Response · Authors · 2023-11-15
>
> **Thank Reviewer 5gUj for the careful and insightful review of our manuscript!**
>
> ## 1. About the "ground truth" carbon footprint numbers and  not "overfitting"?
>
> Within Table 4, we have embraced the "ground truth" carbon footprint values of LLMs as reported by Google (Patterson et al., 2021) and Meta (Wu et al., 2022). It is our contention that these cloud service providers have conducted comprehensive assessments of the operational energy consumption associated with their hardware infrastructures during LLM training. Subsequently, they have employed specific equations, such as Equation 10 and Equation 11 presented in this study, to derive estimates for the operational carbon footprint values. In contrast, we introduce Equations 1 through 9, which serve as a direct means of modeling the operational energy consumption of their computational devices during LLM training. We have then employed their publicly available data to validate the predictions generated by our equations. In the context of Table 4, we have utilized the same LLMCarbon framework to forecast the operational carbon footprint of various LLMs. The results of our validation process reveal a maximum difference of +8.2% between the predicted carbon footprint and the previously published data.
>
>
> ## 2. It is difficult to separate the contributions of the paper from the prior works.
>
> With the continuous expansion in the scale of LLMs, the ecological ramifications concerning carbon footprints have gained heightened significance. The distinctive contribution of LLMCarbon resides in its pioneering endeavor, presenting the inaugural tool capable of meticulously modeling both the operational and embodied carbon footprints associated with an LLM. It is pertinent to emphasize that, heretofore, no prior research has achieved this level of proficiency. While our approach incorporates certain foundational building blocks derived from prior research and relies on previously published data for the validation of LLMCarbon, it is paramount to acknowledge that no prior work has demonstrated the capacity to model the carbon footprint of LLMs, let alone attaining the precision level exhibited by LLMCarbon.
>
> ## 3. There is no measure of utility of an LLM.
>
> We have employed the scaling law (Kaplan et al., 2020) (Hoffmann et al., 2022) to formulate the representation of an LLM's test loss, as exemplified in Equation 3. Within the framework of LLMCarbon, we have the capacity to model scenarios in which the reduction of an LLM's carbon footprint is attainable through an increase in its test loss. As depicted in Figure 8, it is observable that the adoption of a smaller LLM, characterized by a reduced parameter count, can effectively diminish the carbon footprint. Nevertheless, this reduction in carbon footprint is accompanied by an associated increase in the test loss of the smaller LLM. However, now LLMCarbon cannot support the modeling of the accuracy of an LLM adopted in various downstream tasks.
>
> ## 4.	It would be good to discuss the utility of the measurement tool. Is the expectation that model designers pick an appropriate data center, architecture, or training dataset to use based on carbon footprint?
>
> In Section 6 of our work, we have presented a comprehensive analysis encompassing four distinct user case studies, each elucidating the utility of our LLMCarbon tool. These four case studies are as follows:
> 1. Impact of Renewable Energy Adoption in Data Centers: We examine the effect of incorporating renewable energy sources within data centers on the carbon footprint of LLMs.
> 2. Optimal Parallelism Settings in LLM Training: This case study delves into the ramifications of adopting optimal parallelism settings during the training process of LLMs, particularly with regard to their carbon footprint.
> 3. Adoption of New Hardware Accelerators (TPUv3 and TPUv4): The investigation focuses on the implications of integrating advanced hardware accelerators, specifically TPUv3 and TPUv4, across all phases of LLM processing.
> 4. Training Carbon Footprint Scaling Across Various LLMs: We explore the concept of Training Carbon Footprint Scaling across different LLMs, shedding light on the interplay between the carbon footprint, test loss, and LLM parameter count (architecture).
>
> While it is notable that we have not included a user case study explicitly examining the impact of data centers, LLM architectures, and datasets on the carbon footprint of an LLM, it is essential to highlight that our first user case study 1 partially underscores the significance of selecting environmentally sustainable data centers. Moreover, our fourth user case study elucidates the trade-off dynamics involving the LLM's carbon footprint, test loss, and LLM parameter count (architecture). Lastly, our third user case study provides insight into the embodied footprint of an LLM, offering designers the option to make adjustments by choosing between TPUs and GPUs for LLM training.

---

> > ### Comment · Reviewer_5gUj · 2023-11-22
> > **Thanks for the detailed responses**
> >
> > Thank you for answering the weakness points raised in the review.
> >
> > I would still like the answer to the following question:
> > Did you tune your equations or its parameters to reduce the error? If so, how do we know that LLMCarbon is not "overfitting"?
> >
> > For point (2), I understand that you demonstrated superior modeling capacity/proficiency using existing tools. I'm asking what research questions (if any) did the paper address? Are all of the equations well-known, and did prior works estimate similar metrics (such as cost or energy use) for ML architectures using them?
> >
> > I'm satisfied with your responses for the rest of the issues raised.

---

> > > ### Author Response · Authors · 2023-11-23
> > >
> > > ## For point (2), what research questions (if any) did the paper address?
> > >
> > > We present an end-to-end carbon footprint model for LLMs, encompassing several key components: the parameter model, neural scaling model, FLOP model, operational carbon model, hardware efficiency model, and embodied model (illustrated in Figure 1). While the neural scaling model (Kaplan et al., 2020), the PUE aspect of the operational carbon model (Henderson et al., 2020), and the total carbon footprint of the embodied model (Singh et al., 2020) are derived from previous research, we extend and adapt them to suit the requirements of LLMCarbon, introducing Equation 8 and Equation 12 as our original contributions.
> > >
> > > Furthermore, we introduce the parameter model and the FLOP model to address the specific case of mixture-of-experts (MoE) LLMs, introducing Equation 2 and providing further details in Appendix A. Notably, no prior work has ventured into these aspects. A crucial highlight is our development of the hardware efficiency model (depicted in Figure 5, along with Equation 6 and Equation 7) entirely from scratch, leveraging data published in prior works to validate this model. Constructing all these facets of LLMCarbon is an original endeavor, driven by unique motivations and necessities not addressed in prior research.

---

### Official Review · Reviewer_KiRY · 2023-11-01

**Soundness:** 3 good
**Presentation:** 3 good
**Contribution:** 2 fair
**Rating:** 5
**Confidence:** 4

**Summary:**

In this paper, the authors proposed an end-to-end carbon footprint predictor for llm services. Specificall, the predicted carbon footprint is the sum of two parts: operational carbon footprint and embodied carbon footprint. The former one is calculated with a model by taking model parameters, hardware efficiency as input; The latter one is calcuated with another model by taking hardware type, chip area, system power as input. Through extensive comparison experiments, the predicted results shows a better performance with at most 8.2% error.

**Strengths:**

- Compared with baselines, the proposed llmcarbon generalize the carbon prediction to various network architectures (dense llm and moe llm), various hardwares (gpu and tpu) and various phase (training, inference, experimentaiton and storage).

- The key submodels in llmcarbon are elaborated with mathematic formulation and detail description. All the architecture is easy to follow.

**Weaknesses:**

- Some parameter is crucial to the final predicted results, the process of setting parameters should be clarified. For example, in equation 3, \alpha, \beta are the fitting parameters. However, the fitting dataset and fitting method are missing.

- Apart form the parameters discussed in the paper, more factors also affect the footprint. e.g., the paramter precision (fp16, int8, int4) and the implementation of kernel operation.

- There are not any text description about figure 3 and figure 4.

- The proposed llmcarbon is somewhat like a system design with many prior experience from sota, the algorithm contribution are not notable.

**Questions:**

See above weakness part.

---

> ### Author Response · Authors · 2023-11-15
>
> **Thank Reviewer KiRY for the careful and insightful review of our manuscript!**
>
> ## 1. Some parameter is crucial to the final predicted results, the process of setting parameters should be clarified. For example, in equation 3, \alpha, \beta are the fitting parameters. However, the fitting dataset and fitting method are missing.
>
> We have incorporated Equation 3 as presented in (Hoffmann et al., 2022). In the subsection titled "Training Carbon Footprint Scaling" within Section 6, we stated the following: " To compute the test loss, we adopt the fitting constants including α = 0.34, β = 0.28, A = 406.4, B = 410.7, and E = 1.69 for Equation 3 from (Hoffmann et al., 2022). Since the test loss of an MoE LLM with P parameters is similar to that of its dense counterpart with only P/8 parameters (Rajbhandari et al., 2022), we decreased the P of MoE LLMs to P/8 in Equation 3."
>
> ## 2. Apart form the parameters discussed in the paper, more factors also affect the footprint. e.g., the parameter precision (fp16, int8, int4) and the implementation of kernel operation.
>
> Our objective is to establish a baseline framework capable of representing the carbon footprint associated with LLMs. It is noteworthy that, in the present study, we have not taken into account certain sophisticated algorithmic optimizations, including but not limited to speculative decoding, model distillation, and token pruning. Additionally, we have not incorporated advanced hardware enhancements such as aggressive quantization (utilizing formats like fp16, int8, and int4). We intend to explore the integration of these advanced features in our future work.
>
> ## 3. There are not any text description about figure 3 and figure 4.
>
> We have made textual alterations by introducing descriptive elements for Figure 3 and 4, which are delineated in blue text.
>
> ## 4. The proposed LLMCarbon is somewhat like a system design with many prior experience from SOTA, the algorithm contribution are not notable.
>
> As the size of LLMs continues to grow, the environmental impact in terms of carbon footprints becomes increasingly noteworthy. Consequently, the development of a precise tool for modeling the carbon footprints of LLMs is of paramount importance. The principal contribution of LLMCarbon lies in its pioneering capability to accurately characterize both the embodied and operational carbon footprints associated with LLMs. While LLMCarbon leverages established building blocks and incorporates previously published data for validation purposes, it is worth noting that prior research endeavors have not achieved the same level of precision in modeling both the embodied and operational carbon footprints of LLMs.

---

> ### Comment · Reviewer_KiRY · 2023-11-23
>
> Thanks for providing detailed response.
> Overall, I think this work is a good initiative and baseline for carbon footprint in LLM.
> It would be great the author could build a benchmark tool and standards like MLPerf, and open-source it to the community and industrial, since the field is fast iterated, not only the hardware, but also the training and inference techniques.
> The carbon footprint computing logic will be greatly changed based on those newly evolved techniques.
> With such a tool/benchmark to push the industrial moving more green in LLM, I think that is more important than an ICLR publication.

---

### Official Review · Reviewer_aKac · 2023-11-02

**Soundness:** 3 good
**Presentation:** 3 good
**Contribution:** 3 good
**Rating:** 8
**Confidence:** 3

**Summary:**

This paper introduces LLMCarbon, a comprehensive cost model developed to estimate the carbon footprint associated with various stages of LLM computation, such as training, inference, experimentation, and storage. LLMCarbon extends its applicability beyond previously established models by incorporating support for a broader range of LLM architectures, with a specific focus on the mixture-of-experts architecture and more types of hardware. The paper provides an in-depth discussion of LLMCarbon’s cost model, covering aspects like parameter count, FLOPs, and hardware efficiency. To validate LLMCarbon’s efficacy, we present experimental results across different LLMs and hardware configurations.

**Strengths:**

This paper is both well-written and well-motivated, addressing the critical issue of environmental sustainability by studying the carbon footprint of LLM computation. The authors have proposed a cost model that appears to be both fine-grained and carefully designed, reflecting a thorough approach to this significant topic.

**Weaknesses:**

- the procedure of how LLMCarborn figures out the optimal parallelism is not quite clear.
- It is not clear how to use LLMCarborn to guide the design of future generations of LLM architectures and figure AI accelerators.
- The relationships and connections between the proposed hardware efficiency and metrics like arithmetic intensity and MFU are not clear.

**Questions:**

- For figuring out the optimal parallelism strategy using LLMCarborn, how different are the strategies returned by LLMCarborn compared to the one solved using the compilation approach, e.g., the one proposed in [1]?
- Instead of scaling law, recent LLM pre-training efforts usually leverage a certain training number of tokens, e.g., 1T/1.4T in LLaMA. Would it be sufficient to directly use the number of training tokens instead of loss/perplexity?
- How can one use LLMCarborn to guide the designs of future generations of LLM architecture/training procedures or even AI hardware design?

[1] https://www.usenix.org/system/files/osdi22-zheng-lianmin.pdf
[2] https://arxiv.org/abs/2302.13971

---

> ### Author Response · Authors · 2023-11-15
>
> **Thank Reviewer aKac for the careful and insightful review of our manuscript!**
>
> ## 1. The procedure of how LLMCarborn figures out the optimal parallelism is not quite clear. For figuring out the optimal parallelism strategy using LLMCarborn, how different are the strategies returned by LLMCarborn compared to the one solved using the compilation approach, e.g., the one proposed in [1]? [1] https://www.usenix.org/system/files/osdi22-zheng-lianmin.pdf
>
> The primary objective of modeling optimal parallelism in LLMCarbon is to calculate the maximum hardware efficiency for a given combination of hardware and LLM configurations. Hardware efficiency plays a pivotal role in determining both the training duration (Equation 7) and the operational energy (Equation 8) of the system. It's noteworthy that whether LLMs are manually compiled [Narayanan et al., 2021] (Megatron-LM) or automatically compiled [1] on a specific hardware configuration, the maximum hardware efficiency remains constant. The validation of LLMCarbon's optimal parallelism modeling is carried out against the results obtained from manual compilation [Narayanan et al., 2021] (Megatron-LM). As indicated in Figure 7(a) in [1], the throughput achieved through automatic compilation is marginally lower than that achieved through the manual compilation technique (Megatron-LM).
>
> We have revised the procedure explaining how LLMCarbon computes optimal parallelism. Please review the modified sections highlighted in blue in the PDF file. The central concept of optimal parallelism modeling involves the utilization of a polynomial regression model, trained using maximum hardware efficiency data from [Narayanan et al., 2021], to predict the maximum hardware efficiency based on LLM parameters. Each recorded maximum hardware efficiency value corresponds to an optimal number of computing devices (GPUs). Furthermore, LLMCarbon accommodates scenarios involving MoE LLMs and training with a fewer number of computing devices than the optimal configuration dictates.
>
>
> ## 2. Instead of scaling law, recent LLM pre-training efforts usually leverage a certain training number of tokens, e.g.,1T/1.4T in LLaMA. Would it be sufficient to directly use the number of training tokens instead of loss/perplexity?
>
> In brief, no. The scaling law takes into account not only the number of training tokens but also factors in the model's parameter count and the computational resources utilized during training. Relying solely on the number of training tokens allows us to address only a specific LLM architecture characterized by a predetermined parameter count.
>
> ## 3. It is not clear how to use LLMCarborn to guide the design of future generations of LLM architectures and figure Accelerators. How can one use LLMCarborn to guide the designs of future generations of LLM architecture/training procedures or even AI hardware design?
>
> We have presented four user case studies facilitated by LLMCarbon in Section 6. Specifically, as illustrated in Figure 8, designers can evaluate the carbon footprints and test loss of various prospective LLM architectures to determine potential candidates for future development. Furthermore, by referring to Figure 7, hardware designers can ascertain whether a new generation of TPUs will lead to a reduction or increase in the training carbon footprint of state-of-the-art LLMs.
>
> ## 4. The relationships and connections between the proposed hardware efficiency and metrics like arithmetic intensity and MFU are not clear.
>
> When the architecture of an LLM is fixed, the concept of hardware efficiency pertains to the extent of inactivity exhibited by computational devices, such as GPUs, during the training process of the LLM. This measurement allows LLMCarbon to make estimations regarding the duration of the training process (as specified in Equation 7), as well as the mean energy consumption of these computational devices throughout the LLM training (as elucidated in Equation 8), relying on the values of hardware efficiency. On the contrary, on the same hardware platform, the arithmetic intensity and MFU may be determined by the architecture of the LLM. Some architectures may involve higher arithmetic intensity, others may need a lower one. So the hardware efficiency and the arithmetic intensity are two totally different concepts.

---

> > ### Comment · Reviewer_aKac · 2023-11-22
> > **Thank you for your response**
> >
> > I thank the authors for their careful and thorough responses, which have overall addressed my concerns. I would especially like to thank the authors for modifying the optimal parallelism section.
> >
> > I would like to raise my overall rating from 6 to 8.

---

### Comment · Reviewer_12is · 2023-11-23
**To other reviewers: it is unreasonable that _all_ carbon-related factors are accounted for and validated**

From reading the other reviewers, I think that many concerns are valid, but I would like to note that most of these concerns are very difficult to model or verify. I think the work proposes a carbon model of unprecedented accuracy and accounts for many factors. I am not aware of work that has similar complexity. While this work does not have new algorithms or does not account for complex factored (embodied carbon for the accelerators) it does improve considerably on previous models and will be a strong tool for modeling carbon usage for LLMs.

As such, I will keep my score and encourage other reviewers to raise theirs as well.

---

### Meta-Review · Area_Chair_U9tt · 2023-12-17

**Metareview:**

This paper describes a set of techniques for analyzing the carbon footprint of LLMs. The paper is remarkably thorough in both the model and the case studies. The authors were very sophisticated in how they looked at the problem, and I expect this paper to be a popular reference/starting-point for researchers and practitioners looking to better understand the environmental impacts of LLMs (and other models). The reviewers presented a wide variety of scores (5, 5, 6, 8, 10). Personally, I lean towards a 10. Much of the reviewer feedback was on nuances or details that the proposed model didn't fully account for. That's a sign that (a) the authors did a great job such that only these nuances were left and (b) a group of reviewers from throughout the ML community was able to suggest extensions to the model to address future scenarios. No model is going to be perfect, but item (b) indicates to me that this framework is extensible enough that future researchers will be able to build on it. I strongly recommend acceptance and consideration for orals/awards. Reviewer 12is is a noted expert in the field of deep learning efficiency and a tough reviewer, and I weigh their score of a 10 very heavily.

**Justification For Why Not Higher Score:**

N/A

**Justification For Why Not Lower Score:**

There are many holes in the proposed modeling framework, but that's intrinsic in any modeling framework. Reviewers that issued lower scores did so because they many such holes. I think the overall work is remarkable, and it's unfair to expect any modeling framework to be perfect.

---

### Decision · Program_Chairs · 2024-01-16

Accept (oral)